# Potential Submerged Macrophytes to Mitigate Eutrophication in a High-Elevation Tropical Shallow Lake—A Mesocosm Experiment in the Andes

Karen Portilla [1,2,*], Elizabeth Velarde [2], Ellen Decaestecker [1], Franco Teixeira de Mello [3] and Koenraad Muylaert [1]

1   Biology Department, KU Leuven University, E. Sabbelaan 54, 8500 Kortrijk, Belgium
2   Laboratorio LABINAM, Ingeniería en Recursos Naturales Renovables, Universidad Técnica del Norte, Av. 17 de Julio 5-21 y Gral. José María Córdova, Ibarra EC100150, Ecuador
3   Departamento de Ecología y Gestión Ambiental, Centro Universitario Regional del Este, Universidad de la República, Maldonado 20000, Uruguay
*   Correspondence: kmportillac@utn.edu.ec

**Abstract:** Submerged macrophytes promote water clarity in shallow lakes in temperate regions via zooplankton refuge, allelopathy, and nutrient competition with phytoplankton, thereby increasing zooplankton grazing. However, in high-altitude Andean ecosystems, these interactions in shallow lakes have received far less attention. To understand the role of submerged plants in a relatively cold ecosystem (typical for the Andean region), two 100 L experiments were conducted in Yahuarcocha Lake, which has a permanent cyanobacterial bloom. In our first experiment, we evaluated the response of the cyanobacteria bloom to different concentrations of *Egeria densa* (15%, 35%, and 45% PVI). In the second experiment, we investigated the interactions between *E. densa* (35% PVI), zooplankton, and the small-sized fish *Poecilia reticulata* as well as their impacts on phytoplankton. We found a strong reduction in cyanobacteria in the presence of *E. densa*, whereas *P. reticulata* promoted cyanobacteria dominance and zooplankton had a null effect on phytoplankton. Remarkably, the combination of *E. densa*, fish, and zooplankton substantially reduced the algae. Our findings showed that the cyanobacteria bloom decreased in the presence of *E. densa*, thereby increasing the water clarity in the high-elevation eutrophic ecosystem in the Andes. This effect depended on the plant volume inhabited and the small-sized fish biomass.

**Keywords:** submerged macrophytes; eutrophication; allelopathy; cyanobacteria bloom; high-altitude shallow lake

## 1. Introduction

Over the past decades, shallow lakes worldwide have experienced serious eutrophication, which has resulted in a loss of their ecosystem services [1–3]. High nutrient loading has changed the dominance of primary producers from submerged macrophytes to nuisance algae blooms [4,5]. One of the most critical consequences of eutrophication is the emergence of toxic cyanobacterial blooms, which release toxins that may harm aquatic communities and pose a threat to humans [6–8].

Shallow lakes are typically rare in mountain regions except on high-elevation plateaus. The Andes are a series of parallel mountain chains that run along the west coast of South America [9]. Between these mountain chains are the inter-Andean plateaus that harbor shallow lakes. Examples of large shallow lakes in the Andes include the Fuquene, Siscunsi, and Santurbán-Berlín lakes in Colombia [10,11]; the Alalay and Poopó lakes in Bolivia [12,13]; the Pomacocha and Huascacoha lakes in Peru [14]; and Lake Colta in Ecuador [15]. The inter-Andean plateau has been cultivated since pre-Columbian times and is densely populated compared to the tropical lowlands [16]. As a result, many of the shallow lakes on

the inter-Andean plateau have become eutrophic during the past decades and suffer from toxic cyanobacterial blooms (e.g., Juan Amarillo Humedal in Colombia) [17]. Compared to shallow lakes in temperate regions and even low-elevation lakes in the tropics, these high-altitude shallow lakes have been poorly studied. Therefore, it is critical to have a thorough understanding of these ecosystems to manage eutrophication and restore these shallow lakes.

When shallow lakes experience eutrophication, they shift from a clear water state dominated by submerged macrophytes to a turbid state dominated by algal blooms [18]. Due to hysteresis in the relationship between phytoplankton productivity and nutrient concentrations, a reduction in nutrient loading is often insufficient to restore the clear water state. Biomanipulation involves the management of the fish community (through an increase in predatory fish and a reduction in planktivorous fish) to increase zooplankton abundance and facilitate top-down control of phytoplankton [18–21]. These mechanisms enhance water clarity, which allows submerged macrophytes to recolonize the lake [19]. Submerged macrophytes can stabilize a clear water state through direct or indirect impacts on phytoplankton. These impacts include providing a diurnal refuge for large zooplankton such as *Daphnia* from fish predation, competition for nutrients and light, and allelopathy [21–25].

Moreover, submerged plants facilitate sedimentation and reduce wind resuspension of lake bed sediments [26]. Some mechanisms through which submerged macrophytes stabilize the clear water state are not operational in tropical and subtropical shallow lakes. In (sub)tropical shallow lakes, the fish community is dominated by small species that forage in macrophyte vegetation, and piscivorous fish are often lacking or present in low abundance [27–31]. As a result, macrophytes do not protect *Daphnia* against fish predation and thus do not enhance top-down control on phytoplankton in (sub)tropical shallow lakes [28,29,32]. Nonetheless, submerged macrophytes may still directly control phytoplankton via nutrient competition and/or allelopathy in tropical and (sub)tropical ecosystems [33–35].

High-elevation shallow lakes in the equatorial Andes are similar to (sub)tropical shallow lakes in that they lack a clear seasonality [36]. However, the climate in the inter-Andean valleys is much cooler than in the tropical lowlands, and temperatures are more comparable to the summer temperatures of temperate shallow lakes. This combination makes these lakes particularly interesting in terms of their ecosystem functioning. In addition, lakes in the inter-Andean valley are isolated ecosystems, and as a result, they typically have low diversity. This characteristic makes these lakes sensitive to invasions by exotic species (e.g., *Odontesthes bonariensis* in Laguna Alalay-Bolivia [12]). As a result, in these lakes the submerged macrophyte and fish communities are often dominated by introduced species. It is unknown how these exotic species affect the functioning of these ecosystems [12].

This study focused on Yahuarcocha Lake, a shallow lake situated on the inter-Andean plateau in Ecuador at 2200 m elevation close to the equator (00°22′ N). Since the last decade, the lake has been suffering from a perennial phytoplankton bloom dominated by the toxic cyanobacterium *Cylindrospermopsis* [37]. Submerged macrophytes were common in the lake in the past (17% plant volume inhabited in 2014). However, they have disappeared in recent years. The macrophytes that were dominant in Yahuarcocha Lake were all exotic species: *Egeria densa*, *Elodea canadensis*, *Myriophyllum aquaticum*, and *Potamogeton pusillum*. The fish community of the lake is also entirely composed of introduced species that are dominated by *Poecilia reticulata*, *Oreochromis niloticus*, and *Cyprinus carpio*.

In temperate lakes, the addition of piscivorous fish and the removal of planktivorous and benthivorous fish is generally an effective way to achieve biomanipulation [18,23]; this method favors the development of large zooplankton and enhances the water clarity via top-down control [18,38,39]. However, transferring current biological restoration techniques to warm lakes is challenging due to differences in biological interactions between cold temperate and warm (sub)tropical/tropical lakes [18,40]. Recent research has demonstrated that adding submerged macrophytes reduces phytoplankton biomass in subtropical regions [20,33,35]. Although the role of submerged macrophytes in managing eutrophication in temperate shallow lakes (and to a lesser extent in tropical shallow lakes) is quite well understood [18,41,42], their role in high-altitude Andean ecosystems has not been studied.

Aquatic vegetation is vital for shallow lakes, and there is a growing interest in restoring plant communities in shallow ecosystems [34,40,43]. This study aimed to evaluate whether submerged macrophytes could play a role in controlling eutrophication at a high altitude in the Andes and whether they might do so by enhancing the top-down control of zooplankton or via direct effects such as nutrient competition or allelopathy. To evaluate this, two mesocosm experiments were set up. One experiment aimed to evaluate the phytoplankton response to different concentrations of *Egeria densa*. A second experiment aimed to study fish–zooplankton–phytoplankton interactions in the presence and absence of a submerged macrophyte (*Egeria densa*) to evaluate whether macrophytes may enhance the top-down control of phytoplankton by providing zooplankton with a refuge from fish predation.

Our main objectives were to determine: (i) how the phytoplankton bloom in Yahuarcocha Lake would react to the different concentrations of *Egeria densa*; and (ii) how the phytoplankton would respond to the interaction between zooplankton, small-sized fish, and the submerged macrophytes. We tested three main hypotheses: (i) the cyanobacteria would reduce their concentrations in a high-altitude shallow ecosystem in the Andes with the addition of *Egeria densa*; (ii) the addition of herbivorous zooplankton (*Daphnia*) would not be efficient at controlling the cyanobacterial bloom; and (iii) the addition of small-sized fish would promote cyanobacteria growth in the eutrophic shallow lake in the high-altitude ecosystems.

## 2. Materials and Methods

### 2.1. Study Area

The mesocosm experiments were conducted in a field near the shore of Yahuarcocha Lake in Ibarra, Ecuador. The lake is located at an elevation of 2200 m above sea level and has an area of 2.6 km$^2$ [37]. The lake has a mean depth of 4 m and a maximum depth of 7 m. The water temperature averages 21 °C ($\pm$2 °C), which is slightly higher than air temperature (annual average 18 °C). This ecosystem is fed by a small stream that originates in the Páramo grassland at an elevation of 3827 m in the Andes and drains a lower-lying agricultural area. The lake is a popular weekend destination for visitors. Both intensifications of agriculture and the discharge of untreated wastewater from numerous restaurants surrounding the lake are probably responsible for eutrophication in the past decades. According to Van Colen et al. [37], the total nitrogen (TN) and total phosphorus (TP) concentrations in the lake in 2014–2015 were 1900 μg L$^{-1}$ and 56 μg L$^{-1}$, respectively, with 63 μg L$^{-1}$ of dissolved inorganic nitrogen (DIN) and 8 μg L$^{-1}$ of soluble reactive phosphorus (SRP). The nutrient concentrations measured during the current study were higher: the TN concentration was 3900 μg L$^{-1}$ and the TP concentration was 120 μg L$^{-1}$ with 28 μg L$^{-1}$ of PO$_4$ and 400 μg L$^{-1}$ of NO$_3$. The lake is turbid and has a Secchi depth of 0.30 m. The chlorophyll *a* concentration at the time of the experiments was 145 μg L$^{-1}$. Rotifers and small cyclopoid copepods mainly represent the zooplankton community. The only *Daphnia* species that occurs in the lake is *Daphnia pulex*, but the individuals are small (<1 mm) with very low densities (0.05–0.5 ind·L$^{-1}$). Submerged macrophytes were present in the lake up to 2014, but they have almost completely disappeared. All plant species were exotic species that were probably accidentally introduced (*Egeria densa*, *Elodea canadensis*, *Myriophyllum aquaticum*, and *Potamogeton pusillum*). The fish community consists only of alien species and is dominated by the small guppy (*Poecilia reticulata*) and green swordtail (*Xiphophorus helleri*) fish.

### 2.2. Experimental Set Up

We carried out two mesocosm experiments to evaluate the phytoplankton response to different concentrations of *Egeria densa*, fish, and zooplankton addition in a high-altitude shallow lake in the Andes. *Egeria densa* is a non-native invasive and widespread species that is found in most tropical, subtropical, and temperate regions [44,45]. We chose *Egeria densa* due to its previous presence in the lake [37]. Both mesocosm experiments were carried out in brown polyethene tanks (diameter: 50 cm, height: 70 cm) filled with 100 L of water from Yahuarcocha Lake. The mesocosms were placed on a field close to the shore of the lake. The

water was collected 10 m from the shore just below the surface and filtered through a 64 μm nylon mesh to remove the large crustacean zooplankton. Each treatment was replicated three times in both experiments. For both experiments, water was stirred daily using a stick to avoid sedimentation of the phytoplankton. Water from the lake (filtered through a 64 μm mesh) was added daily to the tanks to compensate for evaporative losses (about 4 L per tank daily). Submerged macrophytes were removed before stirring to avoid damage to the macrophytes. The water temperature was monitored in the experiment and remained quite stable at 22 °C, which was comparable to the temperature in the lake at the time of the experiment (21 °C).

The experiments were carried out from August to October. A mobile meteorological station was placed in the experimental area. The air temperature, PAR radiation, precipitation, and wind speed were monitored every hour. The meteorological data showed that the mean daytime (06:00–18:00) PAR radiation during the experiments was 459 μE and ranged from 159 to 641 μE, the air temperature ranged from 10 °C to 24 °C with an average of 17 °C, and the mean wind speed was 0.6 m/s.

Because *Egeria densa* no longer occurs in Yahuarcocha Lake, it was collected from a stand close to the shore of the nearby San Pablo Lake. Healthy shoots that were about 50 cm long were selected and carefully rinsed with tap water. They were kept for 15 min in a weak acetic acid solution in water (0.3%) and washed again with tap water to remove macroinvertebrates (mainly snails) and zooplankton [46]. The *Egeria densa* shoots were grouped in bundles of 6 shoots that were placed in a 50 mL plastic cup filled with sand and that was closed with parafilm. The sand served mainly as a weight to keep the macrophytes submerged. Because *Daphnia* is considered an important species for the top-down control of phytoplankton and the *Daphnia* numbers in Yahuarcocha Lake are very low, we added zooplankton from the nearby San Pablo Lake. The *Daphnia* abundance in San Pablo Lake (1.2 ind·L$^{-1}$) was 3 times higher than in Yahuarcocha Lake (0.4 ind·L$^{-1}$). The zooplankton was collected from the Yahuarcocha and San Pablo lakes using a Schindler-Patalas trap (64 μm) and added to the mesocosms. The zooplankton was collected at a depth of 4 m in both lakes because *Daphnia* tends to avoid the lake surface during the daytime. The equivalent number of zooplankton from 90 L (~144 *Daphnia* individuals per tank) of water from each lake was added to the experiment with zooplankton. For the fish (F) treatments, we collected the small-sized fish *Poecilia reticulata* from Yahuarcocha Lake (natural densities still unknown).

The first experiment was carried out in August 2018 and aimed to evaluate the direct impact of the submerged macrophyte *Egeria densa* on the phytoplankton biomass. We compared four treatments: a control containing no macrophytes (C) and three treatments with varied abundances of *Egeria densa* that included 15%, 35%, and 45% plant volume inhabited (PVI). The control treatment (C) consisted of water from Yahuarcocha Lake that was previously filtered to remove the large zooplankton and contained phytoplankton at the natural concentration. The additional treatments received different amounts of *Egeria densa* corresponding to 2.5 gWW L$^{-1}$ (g of wet weight per liter), 4 gWW L$^{-1}$, and 5 gWW L$^{-1}$ to realize the 15%, 35%, and 45% PVI in the tanks (Figure 1a). The experiment was monitored for 10 days. We collected water samples to analyze the phytoplankton, chlorophyll *a* concentration, cyanobacteria abundance (*Planktothix*), *Planktotrix* filament length (size), and concentrations of total nutrients at the start of the experiment and the end (Table 1).

The second experiment was carried out in October 2018 and aimed to determine whether the macrophyte *Egeria densa* could indirectly enhance the top-down control of phytoplankton by reducing the predation pressure from *Poecilia reticulata* on zooplankton. In this experiment, we compared six treatments. Four treatments did not contain macrophytes: a control treatment without zooplankton and fish (C), a treatment with only zooplankton (Z), a treatment with only fish (F), and a treatment with zooplankton as well as fish (ZF). Two additional treatments contained macrophytes at a PVI of 35%: a treatment with only zooplankton (EZ) and a treatment with zooplankton and fish (EZF). This experiment was monitored for 12 days (Table 1). The control treatment was identical to the previous experi-

ment (C). The second treatment contained an inoculum of large zooplankton (Z), and the third treatment included the fish *Poecilia reticulata* (F). Fish to be added to the F treatments were collected in Yahuarcocha Lake with a 2 mm macroinvertebrate net close to the shore. Only adult *P. reticulata* were selected, and the average size of the fish was 2.4 cm (+/− 0.5 cm; $n = 30$). Ten fish were added per mesocosm tank. All fish survived during the experiment.

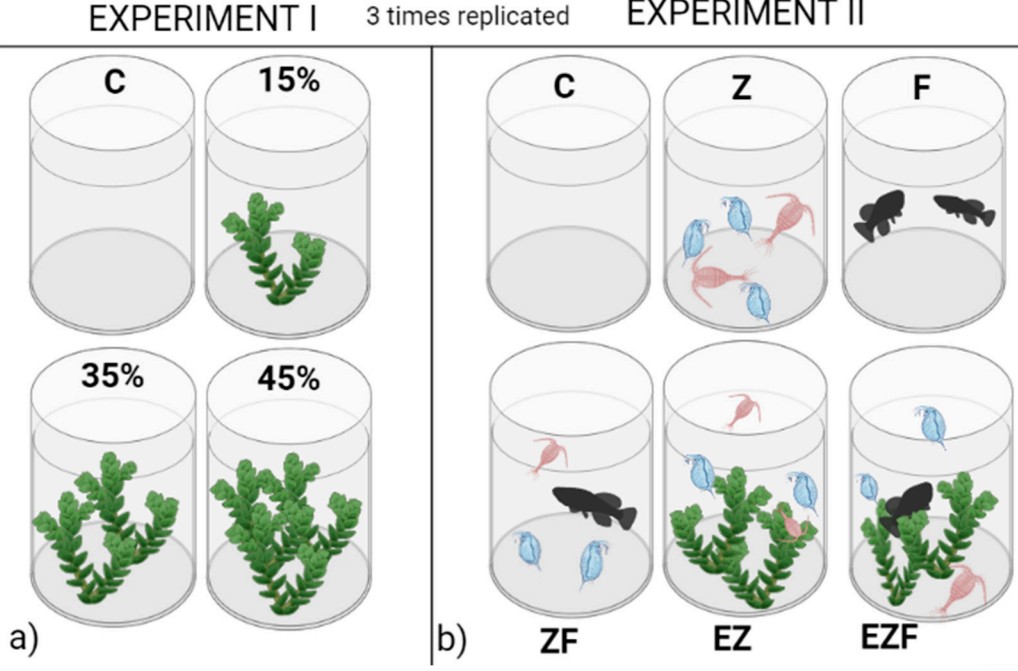

**Figure 1.** (**a**) Experiment I design. Each tank contained 100 L of water from Yahuarcocha Lake. The treatments contained different amounts of fresh *Egeria densa* as follows: "Control" (C), 15% with 2.5 gWWL$^{-1}$ (g of wet weight per liter) of *Egeria densa*, 35% with 4 gWWL$^{-1}$, and 45% with 5 gWWL$^{-1}$. Each treatment was replicated 3 times. (**b**) Experiment II design. Each tank contained 100 L of water from Yahuarcocha Lake. Six experimental units were established: "Control" (C); "Fish" (F); zooplankton (Z), a combination of zooplankton and fish (ZF); a combination of *Egeria densa* (35% with 4 gWWL$^{-1}$) and zooplankton (EZ); and a combination of *Egeria densa*, fish, and zooplankton (EFZ).

**Table 1.** Experimental setup. Experiment I: 4 treatments of C (control) and 15%, 35%, and 45% PVI of *Egeria densa* compared with the initial values (I) of phytoplankton biomass, cyanobacteria density (*Planktothix*), size of *Planktotrix,* and nutrients for a duration of 10 days. Experiment II: 6 treatments compared with initial values of C, Z (zooplankton); F (fish); and the combinations of ZF, EZ (35% PVI of *E. densa* and zooplankton), and EZF for a duration of 12 days.

| Treatments | Variables | | | | |
|---|---|---|---|---|---|
| | Phytoplankton Biomass | *Planktothrix* Density | *Planktothrix* Size | Nutrients (NP) | *Daphnia* Density |
| C | + | + | + | + | − |
| 15% | + | + | + | + | − |
| 35% | + | + | + | + | − |
| 45% | + | + | + | + | − |
| C | + | + | − | + | + |
| Z | + | + | − | + | + |
| F | + | + | − | + | + |
| ZF | + | + | − | + | + |
| EZ | + | + | − | + | + |
| EZF | + | + | − | + | + |

For both mesocosm experiments, we collected samples of water and phytoplankton to evaluate the chlorophyll *a* concentration, nutrients, and phytoplankton community at the start of the experiment and at the end. Samples were taken after stirring the tanks to homogenate the water. For the chlorophyll *a* concentration, samples of 50 mL were filtered through a Whatman GF/F filter and frozen until analysis. For the phytoplankton composition, 50 mL of water was fixed with formalin to a final concentration of 4%. For the *Daphnia* abundance, the entire 100 L of water was filtered through a 64 μm zooplankton net, and a 100 mL concentrate was fixed with formalin at a final concentration of 4%.

### 2.3. Sample Analysis

The chlorophyll *a* was extracted from the filters by soaking the cut-up filter overnight in methanol (99%). The chlorophyll *a* concentration in the methanol extract was determined using a Turner Designs AquaFluor fluorometer that was calibrated against a pure chlorophyll *a* standard. The phytoplankton was counted in a Sedgewick Rafter cell using a light microscope at 100× magnification. At least 200 units or colonies in each sample were counted and identified at the genus level. As *Planktothrix* was the dominant species in all samples (>91% abundance), only the numbers of *Planktothrix* are reported. For the first experiment, the size of at least 100 *Planktothrix* filaments was measured in each sample to monitor changes in the size of this dominant species. For the *Daphnia* density, the entire sample was counted using a dissection microscope, and zooplankton was identified up to the level of genus. Fish were also counted at the end of the experiment.

For total nutrients, 50 mL of water was stored and treated with alkaline persulphate digestion prior to the analysis according to [47]. The TP and TN were measured using automated colorimeter methods with a NOVA 60A photometer equipped with standard kits for phosphate test (Spectroquant 0.010–5.00 mg/L $PO_4$) and nitrate test (Spectroquant 0.2–20.00 mg/L $NO_3$).

### 2.4. Statistical Analyses

We used ANOVA to statistically compare the final values with the initial values of the chlorophyll *a* concentration, nutrients (TN and TP), *Planktothrix* abundance, *Planktothrix* size, and *Daphnia* abundance in the different treatments. In each experiment, we conducted a post hoc analysis by using a Tukey's HSD test. To find the ANOVA assumptions, data were transformed (log10 (x + 1)) when it was necessary. All statistical analyses were performed using R version 3.5.2 (The R Foundation for Statistical Computing, Vienna, Austria).

## 3. Results

### 3.1. Experiment 1

Yahuarcocha Lake presented a high value of chlorophyll *a* concentration at the beginning of the first experiment (139 μg $L^{-1}$). The phytoplankton biomass as chlorophyll *a* concentration exhibited a significant decline in the presence of *Egeria densa* after 10 days (Figure 2A, Table 2). As we expected, the treatment with the greatest macrophyte density of 45% (5 gWWL$^{-1}$) presented the highest reduction in the chlorophyll *a* concentration (by 75%), resulting in a final value for the chlorophyll *a* of 35 μg $L^{-1}$ (±11.5) (Figure 2A, Table 2). Around 48% of the chlorophyll *a* concentration was reduced in the treatments with 35% *Egeria* (4 gWWL$^{-1}$), showing a final value of 72 μg $L^{-1}$ (±17.28) of chlorophyll *a* and 106 μg $L^{-1}$ (±12) in the treatment with 15% *Egeria* (2.5 gWWL$^{-1}$) (Figure 2A, Table 2). No significant increase or decrease in the chlorophyll *a* concentration was observed in the control treatment (C) (see Table A1). The cyanobacterium *Planktothrix* presented a significant decline in abundance by up to −93% with the treatment with the highest amount of *Egeria* (45% PVI) and −60% for the treatment of 35% PVI as shown in Figure 2B. No significant changes were observed in the C and 15% PVI *Egeria* treatments (Figure 2B, Table 2).

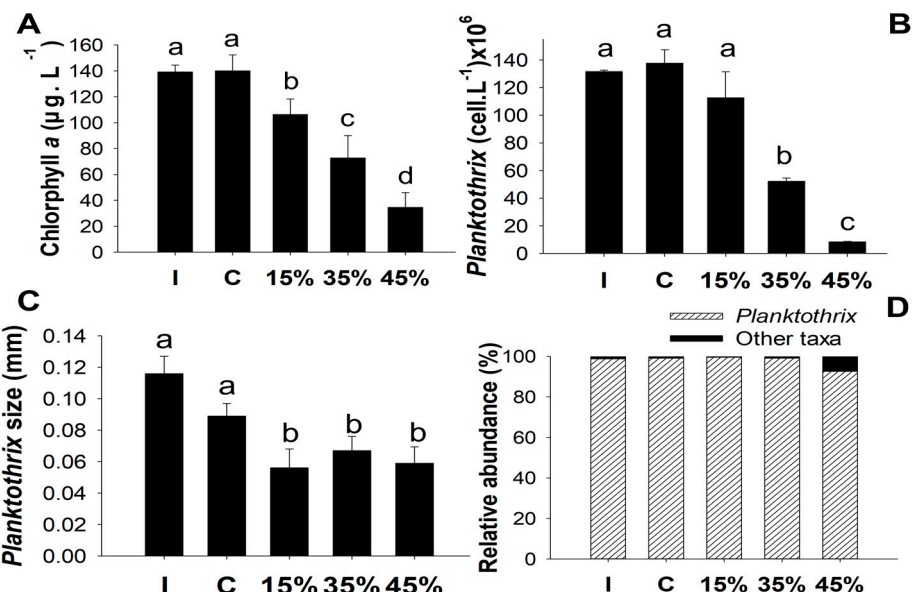

**Figure 2.** Mean values of chlorophyll *a* concentration (±standard error) in the different treatments corresponding to 15% PVI or 2.5 gWWL$^{-1}$ (g of wet weight per liter), 35% PVI (4 gWWL$^{-1}$), and 45% PVI (5 gWWL$^{-1}$) (**A**), *Planktothrix* cell per liter (**B**), *Planktothrix* size (**C**), and relative abundance of phytoplankton community (**D**).

**Table 2.** Results of ANOVA analysis. Experiment I compared the chlorophyll *a*, *Planktothrix* density and size, TN, and TP based on the factors of different percentages of PVI; Experiment II compared the chlorophyll *a*, *Planktothrix* density, TN, TP, and *Daphnia* density based on the factors of different percentages of macrophytes, zooplankton, and fish.

| | Experiment I | | | Experiment II | | |
|---|---|---|---|---|---|---|
| | df | F | P | df | F | P |
| Chlorophyll *a* | 4 | 106.6 | <0.001 | 6 | 23.67 | <0.001 |
| *Planktothrix* density | 4 | 47.94 | <0.001 | 6 | 34.15 | <0.001 |
| *Planktothrix* size | 4 | 18.07 | <0.001 | - | - | - |
| TN | 4 | 1.32 | 0.333 | 6 | 1.5 | 0.25 |
| TP | 4 | 0.75 | 0.575 | 6 | 0.77 | 0.6 |
| *Daphnia* density | - | - | - | 4 | 45.91 | <0.001 |

We observed variations in the filament length of *Planktothrix*. We found a reduction in size in the *Egeria* treatments compared to the C treatment (Figure 2C). The cyanobacterium *Planktothrix* dominated the phytoplankton community with a 99% total abundance at the start of the experiment. After 10 days, this species remained dominant in all treatments (92–99%) (Figure 2D). Nevertheless, the chlorophytes slightly increased (7%) in the 45% *Egeria* treatment (Figure 2D).

High total nutrient concentrations were observed at the start of the experiment: 89 µg L$^{-1}$ (±21) for TP and 4866 µg L$^{-1}$ (±750) for TN (Table A2). However, no significant changes were observed for TP or TN at the end of the experiments.

### 3.2. Experiment 2

The second experiment aimed to elucidate the influence of the submerged macrophyte *Egeria densa* on the interactions between phytoplankton, zooplankton, and fish. The initial chlorophyll *a* concentration at the start of the experiment was lower than in the first experiment at 73 µg L$^{-1}$ (±9) (Figure 3A, Table 2). The chlorophyll *a* concentration did not increase significantly during the course of 12 days in the control (C) treatment. The addition of large zooplankton (Z) to the experiment did not result in a decrease in the phytoplankton because

the chlorophyll *a* concentration did not differ significantly from the control (C) treatment with 84 μg L$^{-1}$ (±9) (Figure 3A, Table 2). The chlorophyll *a* concentration was significantly higher in the treatments with fish (F) and fish plus zooplankton (ZF), increasing 27% for the F treatment with 92 μg L$^{-1}$ (±7) and 25% for the ZF treatment with 91 μg L$^{-1}$ (±7) (Figure 3A, Table 2).

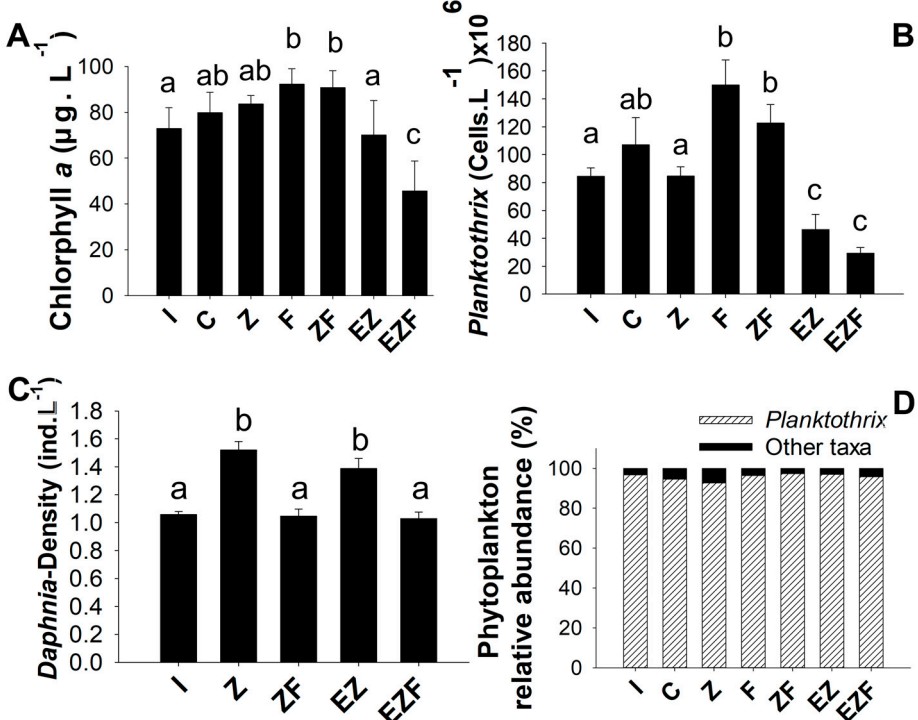

**Figure 3.** Mean values of chlorophyll *a* concentration (±standard error) (**A**) in the different treatments: initial values (I) in comparison with the treatments of control (C), zooplankton (Z), fish (F), zooplankton + fish (ZF); *Egeria densa* + zooplankton (EZ), and *Egeria densa* + zooplankton + fish (EZF); *Planktothrix* density (**B**); *Daphnia* density (**C**); and relative abundance of phytoplankton community (**D**).

In the treatment with *Egeria densa* (35% PVI) in combination with zooplankton (EZ), the chlorophyll *a* concentration did not differ from the control treatment with 70 μg L$^{-1}$ (±15) (Figure 3A, Table 2). In contrast, when *Egeria densa* was added to zooplankton and fish (EZF), a significant decrease of 37% with 46 μg L$^{-1}$ (±13) in the phytoplankton biomass concentration was observed compared to the control treatment (Figure 3A, Table 2).

*Planktothrix* (cell. L$^{-1}$) abundance remained constant in the control treatment (C) as well as in the treatment with zooplankton (Z). However, a significant increase in the *Planktothrix* abundance was observed in the treatments in which fish were added either alone (F) or in combination with zooplankton (ZF) as shown in Figure 3B and Table 2. The treatments containing *Egeria densa* (35% PVI), on the other hand, had a significantly lower *Planktothrix* abundance with a reduction of 45% for the EZ treatment and 65% for the EZF treatment (Figure 3B, Table 2). Even with the addition of fish, the *Planktothrix* density significantly declined in combination with *Egeria densa* (35% PVI). Nevertheless, fish (F) alone significantly increased the *Planktothrix* biomass over the course of the experiment.

At the beginning of the experiments, the *Daphnia* abundance (from the inoculum) was 1.2 ind. L$^{-1}$ in the treatments that received a zooplankton inoculum from the Yahuarcocha and San Pablo lakes (Z, ZF, EZ, EZF). In the absence of fish in the Z and EZ treatments, the *Daphnia* abundance increased significantly: 35% for Z and 30% for EZ (Figure 3C, Table 2). There was no significant change in *Daphnia* abundance in the treatment in which fish were added whether *Egeria densa* was present (EZF) or not (ZF) (Figure 3, Table 2).

After 12 days, the cyanobacteria *Planktothrix* continued to dominate the phytoplankton's relative abundance in all treatments (>92%) (Figure 3D). The total nutrient concentrations did not differ significantly between the treatments at the end of the experiment compared to the initial values (Table A3).

## 4. Discussion

Submerged macrophytes play a significant role in the structuring and functioning of tropical, (sub)tropical, and temperate shallow lakes. They stabilize a clear water state through direct and indirect negative impacts on phytoplankton; i.e., by offering diurnal refuge to large zooplankton against fish predation, competition for nutrients, and allelopathy [21–24,41,43]. According to our findings, the phytoplankton abundance declined substantially in the presence of submerged macrophytes; however, these effects were influenced by the percentage of plant volume inhabited rather than by grazing by zooplankton.

### 4.1. Phytoplankton Biomass and Total Nutrients with Different Concentrations of Macrophytes

Our first experimental results showed that the chlorophyll *a* concentration declined significantly in the presence of *Egeria densa* (15–45% PVI). In addition, the cyanobacteria *Planktothrix* showed a significant decrease but at higher concentrations of submerged macrophytes (35% and 45% PVI). This reduction in the phytoplankton occurred in the absence of large filter-feeding zooplankton. Although no significant changes were observed for TP or TN at the end of the experiments, the higher values in the error bars of the results may have obscured the fact that *Egeria densa* did not significantly reduce the total nutrient concentrations in this experiment.

The biological control of phytoplankton via submerged macrophytes can be achieved by exploitative competition for resources and allelopathic interference competition [22,48–50]. It has been reported that *Egeria densa* reduced nutrient concentrations in the water column in subtropical lakes in Uruguay [51] and outdoor experiments in Florida [52]. Based on our results, it was clear that the phytoplankton declined in the presence of *Egeria densa*, which suggested that possibly two mechanisms accounted for this reduction in allelopathy and nutrient competition. Similar observations of phytoplankton reduction in the presence of *Egeria densa* were attributed to allelopathy and nutrient competition by Vandestukken et al. [33], who used the submerged macrophytes *Egeria densa* and *Potamogeton illinoensis* to control phytoplankton growth in a mesocosm experiment in Uruguay. Previous research suggested that *Egeria densa* produce allelochemicals that reduce the growth of other species, including cyanobacteria [45,49,53,54]; however, no evidence of allelochemical compounds has been reported for this species. In general, allelochemicals are released by macrophytes in response to the presence of cyanobacteria in the environment, and these allelochemicals in turn inhibit cyanobacteria growth [54,55]. In line with our first hypothesis, although *Planktotrix* tolerated *Egeria densa* at low concentrations (15% PVI), this cyanobacterium quite reduced their concentrations in the presence of *Egeria densa* at higher concentrations. Similar results were shown in the study by Amorim [35] in which the cyanobacteria biomass composed mainly of *Mycrosistis* and *Raphidiphsis raciborskii* showed a significant reduction in the presence of the submerged macrophyte *Ceratophyllum demersum*. Submerged macrophytes are thought to reduce toxic and nontoxic phytoplankton biomass through allelopathy and nutrient competitions in tropical and subtropical shallow lakes [33–35]. We not only observed a reduction in the *Planktothrix* abundance, but also a reduction in the filament length. The changes in the length of *Planktotrix* have been documented when *Planktotrix* was exposed to changes such as increased salinity concentrations [56].

Submerged macrophytes can significantly reduce the portion of the light that enters the water column, thus decreasing the phytoplankton abundance [57]. Vandestukken et al. [33] did not find a shading effect of macrophytes on phytoplankton at 35% PVI in the mesocosm. Similar values for macrophytes were tested in our study (from 15% to 45% of the mesocosm volume). According to Mulderij et al. [57], the shading effect is significant when macrophytes occupy a large part of the water column. This effect also depends on the macrophyte species;

for instance, *Ceratofphyllum, Elodea, Myriphyllum,* or *Potamogeto* probably have an intermediate shading effect.

### 4.2. Effects of Zooplankton on Phytoplankton

In the second experiment, we observed that the addition of large zooplankton did not reduce the amount of phytoplankton. Zooplankton in tropical and (sub)tropical areas is commonly dominated by small individuals, while *Daphnia* is rare or present in very low densities as a result of high predation by small omnivorous fish, thereby reducing their grazing pressure on algae [28,29,32]. This reduction contributes to the growth of algae and consequent cyanobacterial blooms [58]. In concordance with our second hypothesis, our study showed that *Daphnia* was not efficient in reducing the *Plankthotrix* biomass and therefore did not enhance the top-down control on the phytoplankton. In this way, *Daphnia* searched for palatable and nutritional algae that was avoiding cyanobacteria [59]. In contrast to our results, Ferrão-Filho et al. [60] observed that the addition of high densities of *Daphnia* (20 ind·L$^{-1}$) drastically reduced toxic cyanobacteria after 17 days in a tropical reservoir in Brazil. The *Daphnia* densities in our experiments were higher than the natural abundance in Yahuarcocha Lake but still low (1.6 ind·L$^{-1}$). Although the abundance of *Daphnia* increased during the course of the experiment in the absence of fish predation, this increase was limited. The limited increase may have been due to a combination of the short duration of the experiment (12 days) and the slow quality of the phytoplankton food, which consisted mainly of cyanobacteria. At these very low densities, the *Daphnia* may not have reduced the cyanobacteria biomass in the eutrophic system. This finding was also reported by the authors of [35], who observed that a low abundance of zooplankton failed to control the cyanobacteria biomass.

### 4.3. Effects of Fish on Zooplankton and Phytoplankton

The fish community in Yahuarcocha Lake consists only of invasive omnivorous fish such as *Oreochromis niloticus, Cyprinus carpio, Xiphophorus hellerii,* and *Poecilia reticulata,* with the last species being the most abundant in Yahuarcocha Lake at the time of the experiments. This is an opportunistic omnivorous species that is capable of eating anything that comes its way [61]. *Poecilia reticulata* can exert intense predation pressure on large-bodied zooplankton, thereby reducing their abundance and indirectly contributing to a high phytoplankton biomass [62]. Small-sized omnivores in general lead to a reduction in cladocerans [63]. Indeed, the addition of fish prevented an increase in zooplankton abundance in our experiment.

In line with our third hypothesis, fish significantly increased the cyanobacterial abundance in treatments free of submerged macrophytes. This was certainly not due to increased predation on the zooplankton because this increase occurred in treatments with or without zooplankton. Some studies indicated that fish may be a source of nutrients that enhance phytoplankton production and favor cyanobacteria [63,64]. In addition, omnivorous fish may promote a strong energy flux from littoral and benthic zones to the pelagic zone, thereby increasing nutrient availability to pelagic primary producers [28,64].

### 4.4. Effects of Fish-Zooplankton and Macrophytes on Phytoplankton

The addition of *Egeria densa* (35% PVI) to the omnivorous fish (*Poecilia reticulata*) and large zooplankton (EFZ) was more effective in cyanobacteria reduction than the isolated addition of zooplankton or fish. In temperate lakes, macrophytes provide refuge to large zooplankton against fish predation, thereby enhancing top-down control of the phytoplankton [65]. This interaction between macrophytes–fish–zooplankton is different in tropical [66] and subtropical ecosystems [28,64,67] in which small omnivorous/planktivorous fish are attracted to submerged macrophytes. In tropical and (sub)tropical shallow lakes, small abundant fish are subject to considerable predation by large fish and aerial predators [68,69]. Therefore, they use submerged vegetation as protection from predators. As a result, zooplankton is not safe from predation among submerged macrophytes. Indeed, we observed

no higher zooplankton abundance in mesocosms with fish when *Egeria densa* was present than when *Egeria densa* was absent.

In the second experiment, the *Planktothrix* abundance declined in the treatments with *Egeria densa*; however, the reduction effect was not as strong as in the first experiment. Moreover, concerning the chlorophyll *a* concentration, a significant reduction was only observed in the mesocosms containing fish and macrophytes. A possible explanation for these observations is that the *Egeria densa* plants were in a poorer condition than in the first experiment. This might have been due to overgrowth by periphyton. An increase in nutrient availability benefits epiphytes and this in turn negatively impacts the submerged vegetation; these epiphytes seem to be less sensitive to macrophyte allelochemicals than phytoplankton [70]. It has been reported that small-sized fish can feed on periphytic algae on the surface of the leaves of macrophytes [68], which reduces periphyton from macrophytes. According to Meerhoff et al. [28], the overall lower periphyton biomass on subtropical plants may be due to fish feeding activity on periphyton. Predation by fish on periphyton possibly may have boosted the health of *Egeria densa* in the mesocosms, which in turn resulted in a stronger negative impact of the macrophyte on the phytoplankton chlorophyll *a* concentration.

## 5. Conclusions

Our findings have significant implications for lake restorations at high altitudes in the Andes. We showed that the cyanobacterial bloom concentrations were reduced in the presence of the common submerged macrophyte *Egeria densa* in a relatively cool shallow lake at a high elevation in the Andes. Our results suggested that introducing submerged macrophytes in eutrophic conditions can improve water clarity. However, these factors were influenced by the plant volume inhabited and the fish biomass. Our results did not allow us to draw conclusions regarding whether the reduction in cyanobacteria in the presence of the submerged macrophytes was due to allelopathy, nutrient competition (dissolved nutrients), or other mechanisms. The investigations of the role of submerged macrophytes are increasing in temperate and warm regions and started in high-altitude areas. The knowledge of the role of aquatic plants in the structure, trophic dynamic, and functioning of shallow ecosystems gained from temperate and warm climates will be significant for lake restoration of eutrophic ecosystems in relatively cold eutrophic lakes in the Andes. To our knowledge, this study provided the first evidence of a decline in cyanobacteria abundance in presence of submerged macrophytes in a relatively cold ecosystem at a high altitude in the Andes. We encourage the testing of artificial and natural submerged plants and their implications in distinguishing structural effects and allelopathic effects on phytoplankton biomass. Future investigations in this type of ecosystem should take into account the fish abundance and fish removal.

Yahuarcocha Lake is a eutrophic ecosystem with perennial cyanobacterial blooms. The lake is completely dominated by exotic species. Submerged macrophytes disappeared some years ago, most likely due to an increase in the lake's turbidity. Cyanobacteria blooms have become a common scenario in fish kill events. There is a need to tackle eutrophication in this ecosystem, and the recolonization of submerged plants is required. However, this may be a difficult challenge due to the high turbidity of the water column, the presence of small-sized fish, and the constant nutrient inputs. Fish predation pressure and the constant release of available nutrients are beneficial to algae and periphyton, which increase the water turbidity in lakes.

**Author Contributions:** Conceptualization, K.P., K.M., E.V., and F.T.d.M.; methodology, K.P., K.M., and F.T.d.M.; validation, K.P., K.M., E.V., E.D., and F.T.d.M.; formal analysis, K.P., K.M., E.D., and F.T.d.M.; investigation, K.P., K.M., E.D., and E.V.; resources, K.M. and E.V.; writing—original draft preparation, K.P., K.M., E.V., E.D., and F.T.d.M.; writing—review and editing, K.P., K.M., E.V., E.D., and F.T.d.M.; supervision, K.M. and F.T.d.M.; project administration, K.M. and E.V.; funding acquisition, K.M. and E.V. All authors have read and agreed to the published version of the manuscript.

**Funding:** This research was funded by the VLIR-UOS TEAM project ZEIN2015PR404.

**Data Availability Statement:** Not applicable.

**Acknowledgments:** We thank the students of the Laboratorio de Investigaciones Ambientales (LABI-NAM) team from Universidad Técnica del Norte for their help with the experiments: Jorge Revelo, Andrés Ipiales, Alexis Galarza, Kevin Patiño, Fernanda Benavidez, Erika Pujota, and Diego Olivo. F.T.M. was supported by the Sistema Nacional de Investigadores (SNI) and the Programa de Desarrollo de las Ciencias Básicas (PEDECIBA Geociencias and Biología, Uruguay).

**Conflicts of Interest:** The authors declare no conflict of interest. The funders had no role in the design of the study; in the collection, analyses, or interpretation of data; in the writing of the manuscript; or in the decision to publish the results.

## Appendix A

**Table A1.** For Experiment I, the percentages of inhibition of the phytoplankton biomass, *Planktothrix* density, and *Planktothrix* relative abundance in the control (C) and the 15%, 35%, and 45% PVI treatments after 10 days are shown. For Experiment II, the treatments of fish (F), large zooplankton (Z), and their possible interactions with macrophytes compared to the control (C) after 12 days are shown.

| | Experiment I | | | | Experiment II | | | | | |
|---|---|---|---|---|---|---|---|---|---|---|
| | CI | 15% | 35% | 45% | CII | F | Z | ZF | EZ | EZF |
| Phytoplankton biomass | 0.66 ns | −23.6 * | −47.7 *** | −75.2 *** | 9.6 ns | 26.62 *** | 14.76 ns | 24.65 ** | −3.84 ns | −37.34 *** |
| *Planktothrix* density | 4.4 ns | −14.4 * | −60 *** | −93 *** | 26.7 ns | 77.5 *** | 0.15 ns | 45.18 * | −45.11 * | −65.21 *** |
| *Planktothrix* relative abundance | 99.12 | 99.53 | 99.10 | 92.55 | 94.57 | 97.04 | 92.68 30% | 97.28 −10% | 69.93 34 | 95.76 9.1 |
| *Daphnia* | - | - | - | - | | | | | | |

**Table A2.** Phytoplankton biomass, growth rate, Planktothrix density, and Planktothrix size in the Control (C) and the 15%, 35%, and 45% PVI treatments after 10 days.

| | (Chla ug/L) | SD | Growth Rate (u) | SD | *Planktothrix* (Cell/L) × 10$^6$ | SD | *Planktothrix* Size | SD | TP (µg·L$^{-1}$) | SD | TN (µg·L$^{-1}$) | SD |
|---|---|---|---|---|---|---|---|---|---|---|---|---|
| I | 139.14 | 5.24 | | | 131.7 | 0.9 | 0.113 | 0.011 | 89.3 | 21 | 4866 | 750 |
| C | 140.06 | 12.28 | 0.0003 | 0.008 | 137.6 | 9.8 | 0.089 | 0.008 | 99 | 96.1 | 5000 | 1000 |
| 15% | 106.26 | 11.99 | −0.027 | 0.010 | 112.7 | 18.8 | 0.059 | 0.01 | 73.3 | 20.8 | 4066 | 1101 |
| 35% | 72.72 | 17.28 | −0.067 | 0.023 | 52.2 | 2.3 | 0.056 | 0.012 | 41.6 | 34.5 | 5666 | 577 |
| 45% | 34.5 | 11.5 | −0.144 | 0.032 | 8.4 | 0.2 | 0.067 | 0.009 | 52.3 | 15.3 | 3133 | 2759 |

**Table A3.** Phytoplankton biomass, growth rate, *Planktothrix* density, and *Daphnia* density in the treatments of fish (F) and large zooplankton (Z) after 12 days.

| | (Chla ug/L) | SD | Growth Rate (u) | SD | *Planktothrix* (Cell/L) × 10$^6$ | SD | TP (µg·L$^{-1}$) | SD | TN (µg·L$^{-1}$) | SD | *Daphnia* (ind L$^{-1}$) | SD |
|---|---|---|---|---|---|---|---|---|---|---|---|---|
| I | 72.8 | 9.1 | | | 84.4 | 6 | 75 | 21.2 | 4250 | 353.55 | 1.164 | 0.048 |
| C | 79.8 | 8.9 | 0.006 | 0.008 | 107 | 19.4 | 75.6 | 46.6 | 5066 | 1006.64 | | |
| Z | 83.6 | 3.7 | 0.018 | 0.006 | 149.9 | 17.8 | 133.3 | 66.5 | 5233. | 251.66 | 1.52 | 0.060 |
| F | 92.2 | 6.7 | 0.009 | 0.003 | 84.5 | 6.6 | 116.6 | 47.2 | 5066 | 115.47 | | |
| ZF | 90.8 | 7.3 | 0.015 | 0.006 | 122.6 | 13.2 | 123.3 | 61.1 | 5666 | 577.35 | 1.04 | 0.051 |
| EZ | 70 | 15 | −0.004 | 0.014 | 46.3 | 10.7 | 93.3 | 15.2 | 5166 | 57.735 | 1.56 | 0.07 |
| EZF | 45.6 | 13 | −0.036 | 0.022 | 29.3 | 3.9 | 79.3 | 35.7 | 4800 | 692.82 | 1.27 | 0.02 |

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
