# Peer review of "Potential Submerged Macrophytes to Mitigate Eutrophication in a High-Elevation Tropical Shallow Lake—A Mesocosm Experiment in the Andes"

_water, doi:10.3390/w15010075_

Round 1
Reviewer 1 Report
This manuscript details the methods and results of two experiments intended to address two questions: 1) will Egeria densa control cyanobacteria bloom and which are the causes allelopathy or nutrient competition; 2) how phytoplankton will respond to the interaction between zooplankton, small-sized fish and submerged macrophytes. In addition, the outline three hypotheses: 1) Egeria densa control toxic cyanobacteria bloom through allelopathy and nutrient competition in eutrophic shallow lakes at high altitude in the Andes; 2) the isolates addition of herbivorous zooplankton (Daphnia) is less efficient at controlling cyanobacterial bloom; 3) small-sized fish addition promote cyanobacteria growth in eutrophic shallow lakes in high altitude ecosystems.
The two experiments were conducted in 100 L containers over two different periods approximately 3 months apart. The authors analyzed several response variables and two environmental variables, specifically TP and TN.
This study is an addition to a growing body of research that developed around the idea that aquatic macrophytes will help mitigate phytoplankton blooms, particularly cyanobacteria, in nutrient-rich lakes. In this case, the authors evaluate the relationship using water from a high altitude Andean lake.
I find these questions to be interesting and the study in general would contribute to our understanding of the relationship between phytoplankton density and the presence of certain aquatic macrophytes, particularly in a such a unique ecosystem. However, I encourage the authors to reevaluate some of their assertions, particularly some points where they imply cause and effect, which I feel the data don’t quite support. Most notable is the suggestion that “E. densa control cyanobacteria bloom through allelopathy”, yet there is no supporting information such as reported concentrations of allelochemicals for the study or from the literature that confirms that E. densa produces allelochemicals. Having either would be very useful, providing substantial support for their conclusion.
In addition, I suggest the authors rethink some environmental parameters. For example, I am surprised the authors did not measure light in their experimental chambers. Using brown containers likely reduced light entering from the sides, then adding various densities of E. densa to the containers could have altered light conditions in the treatments. Regarding phosphorus, I am surprised that the authors did not include SRP as a parameter. When I look at the treads for phytoplankton chlorophyll a and density, it appears that TP follows the same trends. That is, as phytoplankton decreases, TP decreases which is not a surprise. But, to determine if E. densa is removing phosphorus from the water, it would have been very useful if the authors had measured phosphorus in solution (e.g., SRP) since that is the fraction that E. densa would have used.
I have added some move specific comments below.
Line 159: Schindler-Patalas trap.
Line 162 to 168: This paragraph seems to be unnecessary since section 2.2 provides a detailed methods section.
Line 170: 2.2 Experiments – I encourage the authors to include the number of replicates for each treatment. I had to really work to find that information.
Line 238: “colonies identified until”, change “until” to “to”; identified to genus level.
Line 241: Daphnia biomass – it appears that the authors actually determined density not biomass.
Results
Line 266: It is generally accepted that chlorophyll a is not a measure of biomass but rather, just a measure of chlorophyll a concentration.
Line 275: I may be missing something mathematically, but is it possible to have a population decline by more than 100%?
Fig 2B: Was Planktothrix counted as cells or as filamentous units?
Line 283-284: From my experience, filamentous cyanobacteria generally do not shrink in size, but rather, the filaments become weakened where a cell has died. Then, the filaments break into two halves. Given that the smaller sizes reported are about ½ of the original filament size, I suspect that is what might be happening. That suggests that the filamentous may have been stressed.
Line 310: should the word “rinsing” be rising?
Line 326: “cells L-1” or units L-1?
Line 337: “Egeria densa did not improve the growth of …..”. This is an example to suggesting a cause and effect when I don’t believe the data support this kind of linkage. I feel that it would be more appropriate to state that “The growth (should growth be density?) of large zooplankton did not increase in the Egeria densa treatment”.
Discussion
Line 350: “Hint” should be Hilt.
Line 357: I don’t think that the data clearly support the conclusion that E. densa “controlled” cyanobacteria densities.
Line 358-359: I encourage the authors to be cautious with the point. Because they only measure TP and not SRP, I wonder if E. densa did reduce soluble phosphorus, but non-soluble phosphorus remained high, and thus there was no significant difference.
Line 364: I encourage the authors to provide additional evidence (e.g., references) supporting this conclusion. Is there literature that indicates that E. densa produces allelochemicals. How about shading in the E. densa treatments?
References
Ref. 38 and 39 are duplicate.
There are other cases where sentence structure and word usage should be modified to improve clarity
Reviewer 2 Report
The manuscript by Karen Portilla et al. addresses an important and still globally insufficiently understood problem of the allelopathic effect of submerged macrophytes on the phytoplankton of lakes dominated by cyanobacteria. The authors showed that Egeria densa can be used for the restoration of high altitude lakes in the Andes. The experiments were properly planned and performed. Unfortunately, their presentation in the manuscript is very chaotic and full of typos and errors, which makes reading very difficult. Therefore, the manuscript requires very careful checking and correction, and in many places also rewriting.
Many references cited in the text are not included in the References, e.g. Scheffer et al., 1993; Liu et a., 2018; Cooke et al., 2012; Houner, 2012; Hint and Gross, 2008; Amorim et al., 2019; Iglesias et al., 2008; Brönmark et al., 2010; Dussault and Kramer, 1996; Bursk et al., 2002; Meerhoof et al., 2003; Seghers, 1967; Meerhoff et al., 2017; He et al., 2018.
Journal names should be capitalized, e.g. Environmental Pollution, not Environmental pollution. Review References and revise other journal titles.
Some items in References are not cited in the text. They should be removed or cited in the text. This applies to the following item numbers: 7, 13, 16, 25, 45.
A lot of information in the Material and Methods chapter is duplicated, e.g. max. depth (line 125 and 127); 100 L of water (line 148 and 168); treatments in experiment I (in line 164 and 173), etc.
The literature should not be cited in the Conclusions section. The discussion with the literature data is to be in the Discussion chapter, and in the Conclusions only the most important achievements resulting from the research carried out.
Requires clarification:
PVI (plant volume inhabited) in line 163 - What was compared to? Where was 100%? In a lake or in experimental tanks?
When an abbreviation is given for the first time, its explanation must be given. So, C, Z, F, EZ, EF, EFZ must be explained on lines 164-166, not in 186-187.
Line 203. Instead of “number of zooplankton from 90 L of water from each lake” it is better to mention how many individuals have been released into each tank.
Line 208. There is “total of 10 fish were added to the tanks”. For a total of 10, or 10 for each tank?
Lines 218-219. There is: “Water from the lake (filtered over a 64 μm mesh) was added daily to the tanks to compensate for evaporative losses”. Why was distilled water not added which would be equivalent to evaporated water? By adding lake water, an additional amount of phytoplankton was introduced, which could have resulted in greater phytoplankton abundance in the control in relation to the initial amount.
Fig. 1a. Why is there less plants in the tank 45% than in 35%? It should be otherwise.
Line 334. There is “1.2 ind. L-1”. Fig. 3C shows that rather approx. 1.0 ind. / L
Error examples:
Line 25. There should be “… did not differ …” (vide line 365).
Lines 159-160. Should be “Schindler-Patalas trap”, not “Shliner patalas trap”.
Lines 180-181 “the month of” is unnecessary.
Line 254. There is: “Daphnia density (mm)”. Density should be in specimens (or ind.) per L. Unless it's not about the density but the Daphnia length.
Line 263-264. Verb is missing in this sentence.
Line 430. There is “prelogical”. I think it should be "pelagic zone".
Claudia Feijoó (No 11 in References) should be written as Feijoó C. and have No 18.
Reference 32 is incomplete.
Items 37 and 38 have the same date 2007. To make the citation different in the text, add the letters 2007a and 2007b.
Items 38 and 39 are duplicated. One should be removed.
In item 53, the phrase "PEKCAN-HEKIM, Z. E. Y. N. E. P.," should be deleted.
Typos examples:
“Poecialia” instead of “Poecilia” (line 140), “specie which in” instead of “species which is” (line 145),
PO4 and NO3 on lines 247-248 should have numbers in subscript.
Not Table 1 but Table 2 in line 305.
Not Experiment I in line 306 but Experiment II.
Not “rinsing” but „rising” in line 310.
Not „a experiment” but „an experiment” in line 370.
Not “biological” but „biologically” in line 374.
Not “1.6 in/L” but „1.6 ind./L” in line 392.
Line 427. Should be “2012).”
Not “specie” but “species” in line 447.
Line 451. Should be “influenced” instead of „influences”.
Line 454. I think it should be “trophic” not “tropic”.
Round 2
Reviewer 1 Report
Revised - Portilla, K. et al. Potential submerged macrophytes to mitigate …..
As noted in my initial review, this manuscript details the methods and results of two experiments intended to address two questions: 1) will Egeria densa control cyanobacteria bloom and which are the causes allelopathy or nutrient competition; 2) how phytoplankton will respond to the interaction between zooplankton, small-sized fish and submerged macrophytes. In addition, the outline three hypotheses: 1) Egeria densa control potentially toxic cyanobacteria bloom through allelopathy and nutrient competition in eutrophic shallow lakes at high altitude in the Andes; 2) the isolates addition of herbivorous zooplankton (Daphnia) is less efficient at controlling cyanobacterial bloom; 3) small-sized fish addition promote cyanobacteria growth in eutrophic shallow lakes in high altitude ecosystems.
As noted earlier, I find these questions to be interesting, and I truly appreciate the time and effort the authors have invested in this revised version of their manuscript. However, I STILL feel that the authors are trying to promote a cause-and-effect relationship between E. densa abundance and the lower values for their algal response variables.
The first sentence in section 4.1 is still a bit strong and I encourage the authors to soften this a bit. The authors state, “Our first experimental results showed that the submerged macrophyte Egeria densa was efficient in reducing phytoplankton chlorophyll a concentration even at a low abundance” and the statement, “This reduction in phytoplankton BY Egeria densa ….”. Based on the data, I am much more comfortable with statements such as “Phytoplankton chlorophyll a concentrations exhibited a decline in the presence of Egeria densa at low densities, and declined significantly with increasing Egeria densa abundance”.
Line 434 – “From our results, Egeria densa had a negative impact on …..”. Again, I feel this is a bit overstated. I suggest “From our results it is clear that phytoplankton declined in the presence of Egeria densa. Two possible mechanisms accounting for this reduction may be ….”.
Line 436 - My only other comment is related to the supporting information (references) related to allelopathy. Many of the references used in the discussion conclude that allelopathy is one of the mechanisms resulting in a decline in various phytoplankton communities. But the majority of the referenced studies do not report actual allelochemical concentration …. They simply assume that a decline a particular algal metric must be due to allelopathy. The authors do include a few references that report the results of allelopathic chemical analyses but even some of those vague. For example, reference 52 lists Egeria dense from a study by Nakia et al. (1999) but there is no compound identified in table 1, and no chemical analysis reported in Nakia et al. (1999). When I go back to the older Nakai et al. (1996), the authors have not characterized the allelopathic compound(s) obtained through methanol extractions. I suspect that I was a bit vague in my earlier review, so I apologize to the authors, but I am hoping that the authors have references that report concentrations of allelochemicals to confirm that E. densa produces allelochemicals. Then noting those concentration in this manuscript would provide substantial support for their suggestion that allelochemicals are a strong possible explanation.
Author Response
- As noted earlier, I find these questions to be interesting, and I truly appreciate the time and effort the authors have invested in this revised version of their manuscript. However, I STILL feel that the authors are trying to promote a cause-and-effect relationship between E. densa abundance and the lower values for their algal response variables.
- The reviewer is correct. We changed the draft according to this suggestion
- The first sentence in section 4.1 is still a bit strong and I encourage the authors to soften this a bit. The authors state, “Our first experimental results showed that the submerged macrophyte Egeria densa was efficient in reducing phytoplankton chlorophyll a concentration even at a low abundance” and the statement, “This reduction in phytoplankton BYEgeria densa ….”. Based on the data, I am much more comfortable with statements such as “Phytoplankton chlorophyll a concentrations exhibited a decline in the presence of Egeria densa at low densities, and declined significantly with increasing Egeria densa abundance”. These results
- The reviewer is right. We changed the draft according to this suggestion
- Line 434 – “From our results, Egeria densa had a negative impact on …..”. Again, I feel this is a bit overstated. I suggest “From our results it is clear that phytoplankton declined in the presence of Egeria densa. Two possible mechanisms accounting for this reduction may be ….”. 365
- The reviewer is right. We changed the draft according to this suggestion
- Line 436 - My only other comment is related to the supporting information (references) related to allelopathy. Many of the references used in the discussion conclude that allelopathy is one of the mechanisms resulting in a decline in various phytoplankton communities. But the majority of the referenced studies do not report actual allelochemical concentration …. They simply assume that a decline a particular algal metric must be due to allelopathy. The authors do include a few references that report the results of allelopathic chemical analyses but even some of those vague. For example, reference 52 lists Egeria dense from a study by Nakia et al. (1999) but there is no compound identified in table 1, and no chemical analysis reported in Nakia et al. (1999). When I go back to the older Nakai et al. (1996), the authors have not characterized the allelopathic compound(s) obtained through methanol extractions. I suspect that I was a bit vague in my earlier review, so I apologize to the authors, but I am hoping that the authors have references that report concentrations of allelochemicals to confirm that E. densa produces allelochemicals. Then noting those concentration in this manuscript would provide substantial support for their suggestion that allelochemicals are a strong possible explanation.
- The reviewer is correct, we have incorporated this suggestion in the draft.
- We thank you for your suggestions. We learn an important lesson in being careful in interpreting results.

Reviewer 2 Report
The authors did a good job in improving the manuscript. All my comments have been taken into account. The manuscript is ready for publication in its present form.
There are insertions in Chinese in two places in the introduction, but I don't think it's the fault of the authors.
Author Response
- The authors did a good job in improving the manuscript. All my comments have been taken into account. The manuscript is ready for publication in its present form.
- We thank to the reviewer for the comment.
- There are insertions in Chinese in two places in the introduction, but I don't think it's the fault of the authors.
- The reviewer is right, the Chinese insertions are not our fault. Thanks
